# Influence of Temperature Conditions during Growth on Bioactive Compounds and Antioxidant Potential of Wheat and Barley Grasses

**DOI:** 10.3390/foods10112742

**Published:** 2021-11-09

**Authors:** Mohammad Zahirul Islam, Buem-Jun Park, Young-Tack Lee

**Affiliations:** Department of Food Science and Biotechnology, Gachon University, Seongnam 13120, Korea; zahirul@gachon.ac.kr (M.Z.I.); qjawns1217@naver.com (B.-J.P.)

**Keywords:** wheat grass, barley grass, growth temperature, biochemical compounds, antioxidant enzymes

## Abstract

Wheat and barley grasses are freshly sprouted leaves of wheat and barley seeds, and are rich sources of phytochemicals. This study was conducted to investigate the effects of day and night temperatures on the growth, bioactive compounds, and antioxidant potential of wheat and barley grasses. Briefly, each grass was cropped in an organic growing medium at 10/5 °C, 20/15 °C, and 30/25 °C (day/night temperature) in a growth chamber by maintaining specific light (12/12 h light/dark; light intensity 150 µmol photons m^−2^ s^−1^) and humidity (60%) conditions for 8 days. The highest growth parameters (height, weight, and yield) were observed at the 20/15 °C growth conditions in both types of grass. Conversely, the lowest growth parameters were observed at 10/5 °C. However, the low growth temperature of 10/5 °C resulted in increased levels of bioactive compounds (total phenol, total flavonoid, and total vitamin C), antioxidant activities (2,2′-azino-bis (3-ethylbenzothiazoline-6-sulphonic acid and 2,2-Diphenyl-1-picrylhydrazyl radical-scavenging activity)), and antioxidant enzymes (guaiacol peroxidase activity, catalase activity, and glutathione reductase) in both types of grass. Therefore, proper temperature growth conditions of wheat and barley grasses may be a convenient and efficient method to increase bioactive compounds and antioxidant potential in our diet to exploit the related health benefits.

## 1. Introduction

Germination and plant growth are generally influenced by various growth factors, including temperature, time, duration, humidity, light intensity, light wavelength, darkness, carbon dioxide, treatment, ozone, and water supply [1,2,3]. These factors vary according to the growth environment, such as a glasshouse, open field, and hydroponic culture. Specifically, in a glasshouse, soil conditions, day and night temperature, air moisture, soil temperature, supplemental light, growth medium, day length, relative humidity, and controlled atmospheric environment influence the growth of sprouts [3,4,5,6,7].

Meanwhile, in a field, sprout growth mainly depends on the seed’s moisture, soaking, priming, soil moisture content, atmospheric temperature, rainfall, frost, field environment, sowing date, fertilizer, treatment, and water supply [2,8]. In a hydroponic culture, sprout growth primarily depends on hydroponic and atmospheric conditions, which include water and atmospheric temperature, atmospheric humidity, atmospheric light intensity, atmospheric light wavelength, night darkness, water pH, and day-night light ratio [2,8,9,10].

Wheat (*Triticum aestivum* L.) and barley (*Hordeum vulgare* L.) are major cereal crops that are consumed worldwide. Various researchers have investigated barley and wheat seeds under different growth conditions. In particular, different soil and atmospheric temperatures (10–20 °C) have been tested to grow barley [6,11]. Moreover, the seed germination of wheat depends on day-night temperatures. Wheat seeds were grown in different growing media at 5–25 °C, with 45–70% relative humidity [1,7,12].

Wheat and barley grass, which are the freshly sprouted leaves from common wheat and barley, are rich sources of phenolic compounds, flavonoids, vitamins, minerals, enzymes, polysaccharides, chlorophylls, and antioxidants [8,13]. Wheat grass has been found to reduce the risk of colorectal cancer, cardiovascular disease, and type 2 diabetes [14,15], whereas barley grass can enhance immunity, reduce cardiovascular diseases, regulate blood pressure, and has anticancer, antidiabetic, and anti-inflammatory effects [13]. Thus, wheat and barley grasses are often utilized in dried powder or as an extract to produce various processed products, including juices, drinks, teas, capsules, and tablets.

Despite increasing popularity, few studies are available regarding the nutritive and bioactive properties of wheat and barley grass as influenced by various growth conditions. In particular, the day and night growth temperatures may affect the bioactive compounds and antioxidant capacity of wheat and barley grass. Therefore, in this study, these compounds were surveyed under different temperature levels using an organic growing medium. Further, this experiment examined the growth parameters, bioactive compounds, antioxidant enzymes, and antioxidants of wheat and barley grass in order to investigate the effects of different growth temperatures.

## 2. Material and Methods

### 2.1. Growth Treatments for Wheat and Barley Grass

Wheat (*Triticum aestivum* cv. ‘Baegjoongmil’) and barley (*Hordeum vulgare*, cv. ‘Keunalbori’) seeds collected from the National Institute of Crop Science, Korea were used to conduct this experiment. Seeds weighing 50 g were soaked in distilled water (dH_2_O) (seed: dH_2_O ratio of 1:1) for 24 h in a plant growth chamber at 20 °C. Five replicates were conducted for each treatment. Then, the seeds were relocated to the organic growing system in the plant growth chamber (Heuksalim, Chungbuk, Korea). The organic growing system contained 35% coco peat, 25% granite soil, 25% mushroom culture, 7.4% perlite, 5% zeolite, 2.5% vermiculite, and 0.1% guano.

The temperature was maintained at 20/15 °C (day/night) during the germination period (5 d). Then, during the young grass growth period, the growth chamber temperatures (day/night) were adjusted for low (10/5 °C), medium (20/15 °C), and high (30/25 °C) with light (12/12 h light/dark; light intensity 150 µmol photons m^−2^ s^−1^ with 10 W quantum dot light-emitting diodes), at a relative humidity of 60%. After 8 d (13 d after sowing), the 6–15 cm height young grass leaves were collected to measure the bioactive compounds and antioxidant capacities. The enzymes and analysis reagents have been acquired from Sigma-Aldrich, Korea.

### 2.2. Growth Parameters

The growth parameters of the young wheat and barley grasses, including height, weight, and yield, were examined to assess the growth temperature effects.

### 2.3. Chlorophyll Content

The chlorophyll contents of the fresh young wheat and barley grasses (0.10 g) were examined using 5 mL of *N, N*-dimethylformamide (99.8%) for 24 h. The absorbance of the supernatant was measured with an ultraviolet-visible spectrophotometer (Shimadzu, Kyoto, Japan) at 647 nm for chlorophyll b and 664 nm for chlorophyll a, as described by Moran [16], using a 3 mL cuvette.

### 2.4. Preparation of Grass Extract

First, the young wheat and barley grasses were extracted (1:4, *w*/*v*, grass: dH_2_O) at 20 °C. Approximately 10 g (1 cm length) of each raw grass (separately) was ground with a pestle in 40 mL of dH_2_O. The filtered liquid samples were stored at 4 °C until further analysis.

### 2.5. Carotenoid Content

The young wheat and barley grass extracts (0.02 mL) were then mixed with 5 mL of acetone, and the mixtures were incubated under dark conditions at 4 °C for 24 h. The absorbance of the final supernatant was measured using a spectrophotometer at 510 nm.

### 2.6. Bioactive Compound Analysis

#### 2.6.1. Total Phenolic Content

The wheat and barley grass extracts (1 mL) were each mixed with 1 N Folin–Ciocalteu reagent (0.4 mL) and 7% sodium carbonate (2 mL) [17], before incubation and vortexing at room temperature (20 °C) for 20 min. The absorbance of the supernatant was measured using a spectrophotometer at 734 nm. Note that 0–200 ppm gallic acid concentrations (Sigma-Aldrich Co., St. Louis, MO, USA) were used as a standard.

#### 2.6.2. Total Flavonoid Content

The wheat and barley grass extracts (2 mL) were mixed with 10% aluminum chloride (0.1 mL), 1 M potassium acetate (0.1 mL), and dH_2_O (2.8 mL), and were then incubated and vortexed for 45 min at 20 °C [18] to measure the absorbances at 450 nm. For the control, 0.1 mL of dH_2_O was used in lieu of 0.1 mL of aluminum chloride.

#### 2.6.3. Total Vitamin C Content

The wheat and barley grass extracts (2 mL) were each vortexed with 5% (*w*/*v*) metaphosphoric acid (5 mL) for 30 min at 4 °C, and the absorbances were measured at 525 nm.

### 2.7. Antioxidant Enzymes Activity

#### 2.7.1. Guaiacol Peroxidase Activity (EC. 1.11.1.7)

The guaiacol peroxidase activity was measured as described by Putter [19], with a slight modification. Specifically, in the test tube, 3.0 mL of phosphate buffer (0.1 M, pH 7.0), 50 µL of guaiacol solution (20 mM), 100 µL of enzyme sample, and 30 µL of H_2_O_2_ solution (12.3 mM) were added and mixed. The absorbance was then measured at 436 nm. The guaiacol dehydrogenation product (GDHP) formation was measured as a guaiacol peroxidase activity (POD) (GDHP unit/min/mL), in which one unit of enzyme activity is expressed as the amount of enzyme required to oxidize 1.0 µM of GDHP/min/mL:

Unit activity (unit/min/mL) = (absorbance/minute × total volume in mL)/(extraction coefficient × volume of sample taken in mL), where the extraction coefficient is 25 mM^−1^cm^−1^.

#### 2.7.2. Catalase Activity (EC. 1.11.1.6)

The catalase activity (CAT) was measured as described by Aebi [20], with a slight modification. Briefly, 1.7 mL of phosphate buffer (100 mM, pH 7.0), 1.4 mL of H_2_O_2_ solution (150 mM), and 500 µL of enzyme extract were added to a test tube and mixed. The absorbance of the supernatant was measured at 240 nm. Note that one unit of enzyme activity is expressed as the amount of enzyme required to oxidize 1.0 µM of H_2_O_2_/min/mL:

Unit activity (unit/min/mL) = (absorbance/min × total volume in mL)/(extraction coefficient × volume of sample taken in mL), where the extraction coefficient is 6.93 × 10^−3^ mM^−1^cm^−1^.

#### 2.7.3. Glutathione Reductase (EC 1.6.4.2)

The glutathione reductase (GR) was measured according to the method described by Carlberg and Mannervik [21], with a slight modification. Specifically, 1.8 mL of phosphate buffer (50 mM, pH 7.6), 300 µL of ethylene-diamine-tetraacetic acid (EDTA) disodium salt (3.0 mM), 300 µL of nicotinamide adenine dinucleotide phosphate (NADPH, 0.10 mM), 300 µL of glutathione (1.0 mM), and 300 µL of enzyme sample were added to a test tube and mixed. The absorbance of the supernatant was measured at 340 nm. One unit of enzyme activity is expressed as the amount of enzyme required to oxidize 1.0 µM of NADPH/min/g:

Unit activity (units/min/mL) = (absorbance/min × total volume in mL)/(extraction coefficient × volume of sample taken in mL), where the extraction coefficient is 6.22 mM^−1^cm^−1^.

#### 2.7.4. Superoxide Dismutase-like Activity (EC 1.15.1.1)

The wheat and barley grass extracts (0.2 mL) were each mixed with 50 mM Tris-HCl buffer (pH 8.5) (3.0 mL) and pyrogallol (0.2 mL), incubated, and shaken for 30 min under dark conditions for 10 min at 20 °C. Then, the mixtures were reacted with 1 N HCl (1 mL) to terminate the reaction. The absorbances of the supernatants were measured at 420 nm. For the control, dH_2_O (0.2 mL) was used in place of the leaf extract.

### 2.8. Analysis of Antioxidant Activity

#### 2.8.1. 2,2’-Azino-bis (3-ethylbenzothiazoline-6-sulphonic acid) Antioxidant Assay

The half-maximal inhibitory concentration (IC_50_) equivalent ABTS capacity (µg/mL) was measured, as per the method described by Ozgen et al. [22], with slight modifications. Briefly, a radical cation solution (1 mM) was prepared in ethanol and used as a standard. The radical cation (2,2′-azino-bis (3-ethylbenzothiazoline-6-sulphonic acid (ABTS)^+•^)) solution was prepared (7 mM ABTS in water) with potassium persulfate (2.45 mM), and incubated at 20 °C for 12–18 h under dark conditions to achieve a stable oxidative state. Then, the ABTS solution (3 mL) was mixed with the standard (200 µL) or test extract (200 µL), and stored in the dark for 2 h at 20 °C before measuring the absorbance of the supernatant at 734 nm. The absorbance of the ABTS^+•^ control was also measured. α-Tocopherol was used as a positive control.

#### 2.8.2. 2,2-Diphenyl-1-picrylhydrazyl Radical-Scavenging Activity

The wheat and barley grass extracts (1.8 mL) were mixed with 0.4 mM methanol containing 2,2-Diphenyl-1-picrylhydrazyl (DPPH) radicals (1.8 mL), vigorously shaken, and incubated for 10 min under dark conditions before measuring the absorbance at 525 nm [8]. The half-maximal inhibitory concentration (IC_50_) equivalent capacity (µg/mL) of DPPH was calculated. α-Tocopherol was used as a reference.

### 2.9. Statistical Analysis

Using SPSS V.25 (SPSS Inc., Chicago, IL, USA), a one-way analysis of variance (ANOVA) followed by a Tukey test were conducted for the statistical analyses. In addition, the Pearson’s correlation was performed using SPSS software.

## 3. Results and Discussion

### 3.1. Growth Parameters

The growth parameters of the wheat and barley grasses grown at different temperatures are listed in Table 1. The germination rates of the wheat and barley seeds were 95% and 93%, respectively, at 20/15 °C (day/night; data not listed in Table 1), whereas the 10/5 °C and 30/25 °C (day/night) growth temperature conditions hampered the germination rate. After germination (5 days), the growth temperatures were adjusted to 10/5 °C, 20/15 °C, and 30/25 °C (day/night) for 8 days. The grass grown at each temperature setting was collected to measure the height, weight, and yield. Among the treatments, the 20/15 °C grass had the highest height, weight, and yield for both the wheat and barley grasses, which may be because of the optimal water stimulating growth via the pentose-phosphate pathway by the cellular division of apical stem. Meanwhile, the 10/5 °C and 30/25 °C (day/night) growth temperatures resulted in a lower height, weight, and yield, which may be because of a decrease in photosynthesis in the plant. Specifically, low temperatures can cause physicochemical changes that restrict the growth and photosynthesis of seedlings [23], and high temperatures can cause enzymatic changes that limit photosynthesis, thereby reducing plant growth and yield [24]. According to lettuce and maize plant growth [3,25], plant development and yield are greatly reduced by high and low temperatures. Moreover, the wheat grass had higher height, weight, and yield values than the barley grass.

### 3.2. Chlorophyll and Carotenoids Analysis

The chlorophyll and carotenoid contents of the wheat and barley grass extracts are shown in Figure 1. The chlorophyll content was significantly higher in the wheat and barley grass extracts obtained at a growth temperature of 20/15 °C. Further, it was found that the fresh weights were positively correlated with the chlorophyll content in grass extracts. Further, the low and high growth temperatures reduced the chlorophyll contents of the wheat and barley grass extracts. The fixation of carbon dioxide in the Calvin cycle is sensitive to atmospheric stresses, including low and high temperatures [26,27], which can decrease photosynthesis in the plant, resulting in a decreased chlorophyll content [23,24]. Thus, the chlorophyll content most likely increased because of the high rate of chemical change during photosynthesis. At low or high temperatures, plants cannot maintain chloroplast enzymes [28,29], which consequently lowers the chlorophyll content.

Compared with chlorophyll, the carotenoid exhibited the opposite result in both the wheat and barley grasses. At a growth temperature of 10/5 °C, both grass extracts had the highest carotenoid levels (yellow to orange color). A good carotenoid level is desirable because it can contribute to scavenging reactive oxygen species (ROS) [30], which may destroy cell membranes. Compared with the wheat grass, the barley grass had relatively lower total chlorophyll and carotenoid contents, which might be the result of crop type and variety variation.

### 3.3. Bioactive Compounds in Wheat and Barley Grass Extracts

The bioactive compounds in the wheat and barley grasses grown at different temperatures are listed in Table 2. The highest bioactive compound contents, including phenolic, flavonoids, and vitamin C, were observed in the grass extracts obtained at a growth temperature of 10/5 °C. This treatment temperature may enhance the accumulation of bioactive compounds in the grasses by activating the phenylpropanoid pathway, and may improve ROS scavenging. Specifically, the phenolic contents increase because of enhanced phenylalanine ammonia-lyase activities and enzymes responsible for polyphenol biosynthesis in the shikimate pathway [31].

Meanwhile, the vitamin C content probably increased in the germinated wheat and barley as a result of *de novo* synthesis [2,10], and, in turn, contributes to scavenging ROS [30]. At 20/15 °C, the bioactive compounds in both grasses were the lowest. Moreover, wheat grass had higher phenolic, flavonoid, and vitamin C contents than barley grass.

### 3.4. Antioxidant Enzymes in Wheat and Barley Grass Extracts

The antioxidant enzyme contents in the wheat and barley grasses grown at different temperatures are listed in Table 3. The results show that the antioxidant enzyme contents, especially POD, CAT, and GR, were higher in the extracts grown at 10/5 °C than those grown at 20/15 °C. A previous study found that low-temperature-stressed cabbage at 12 °C increased the CAT and GR contents [32], which protect the cells against excess ROS [33]. Thus, under temperature-stressed conditions, the wheat and barley grasses might have developed antioxidant systems, especially enzymatic antioxidants (POD, CAT, and GR) and non-enzymatic antioxidants (phenol, flavonoid, and vitamin C) to minimize the harmful effects of free radicals and ROS. Garratt et al. [34] reported that natural detoxifying enzymes, mainly ascorbate peroxidase (EC 1.11.1.11), catalase (EC 1.11.1.6), glutathione peroxidase (EC 1.11.1.7), and superoxide dismutase (SOD; EC 1.15.1.1), are present in plants. Further, the wheat grass exhibited a higher level of antioxidant enzymes than the barley grass, which may be because of different crops, species, and varieties.

### 3.5. Antioxidant Activities of Wheat and Barley Grass Extracts

The antioxidant activities of the wheat and barley grass extracts are summarized in Table 4. Both grass groups grown at 10/5 °C had the most antioxidant activity in the ABTS and DPPH assays. Meanwhile, a previous study found that at a low temperature (12 °C), the antioxidant activity of cabbage increased because of the decrease in chlorophyll content [32]. In this study, 10/5 °C had the highest antioxidant activity and lowest chlorophyll content. When the chlorophyll content decreases, antioxidant activities may increase in leaves as a result of enzymatic activities [35]. In general, plants develop defense determinants in the presence of atmospheric stress by expanding their enzymatic and antioxidant activities [36], which control ROS levels [9,30]. Herein, there were correlations between ABTS and DPPH for both the wheat (r = 0.924, *p* ≤ 0.01) and barley (r = 0.870, *p* ≤ 0.01) grass extracts, and the antioxidant activity may increase because of the increase in phenolic content in the extracts. Further, wheat grass showed higher levels of antioxidant activity than barley grass in the ABTS and DPPH assays, which may be due to crop variation.

## 4. Conclusions

This study was conducted to investigate the effects of day and night growth temperatures (low, medium, and high) in the organic growing medium on the bioactive compound concentrations, antioxidant enzyme concentrations, and antioxidant activities of wheat and barley grass extracts. Both the low and high growth temperatures caused some level of damage in the wheat and barley grass growth parameters, but triggered some defense mechanisms by increasing the amount of bioactive compounds, antioxidant enzymes, and antioxidant activity. Based on these findings, the proper control of day and night growth temperatures when producing wheat and barley grasses may be a convenient and efficient way to improve the bioactive compound contents and related health benefits.

## Figures and Tables

**Figure 1 foods-10-02742-f001:**
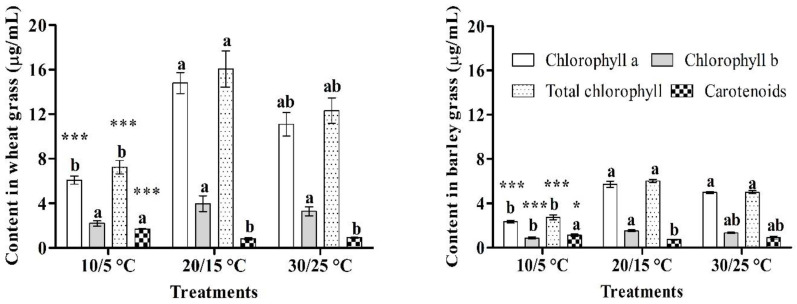
Chlorophyll and carotenoid contents of wheat and barley grass extracts treated at different day and night temperatures. Mean values were analyzed by Tukey test (*n* = 5 ± standard error). NS, *, ***; not significant, and significant at *p* ≤ 0.05 and 0.001, respectively. Values with the same letters in a treatment are not significantly different by Tukey test.

**Table 1 foods-10-02742-t001:** Growth parameters of young wheat and barley grasses treated with different day and night temperatures.

Grass Type	Day and Night Temperature	Grass Height (cm)	Grass Weight (g)	Grass Yield (g/m^2^)
Wheat	10/5 °C	8.32 ± 0.36 b ^z^	76.98 ± 3.06 b	1182.44 ± 47.05 b
	20/15 °C	15.29 ± 0.33 a	113.77 ± 2.89 a	1747.67 ± 44.37 a
	30/25 °C	13.60 ± 0.29 a	92.79 ± 3.18 ab	1425.40 ± 48.91 ab
	*p* value	***	***	***
Barley	10/5 °C	6.01 ± 0.32 c ^z^	59.66 ± 1.99 b	916.44 ± 30.62 b
	20/15 °C	13.97 ± 0.25 a	105.21 ± 2.54 a	1616.13 ± 39.09 a
	30/25 °C	11.52 ± 0.32 b	88.63 ± 3.79 a	1361.44 ± 58.23 a
	*p* value	***	***	***

^z^ Mean separation within columns according to the Tukey test (*n* = 5 ± standard error). *** indicates significance at *p* ≤ 0.001. Values with the same letters in a column are not significantly different by Tukey test.

**Table 2 foods-10-02742-t002:** Bioactive compound contents of wheat and barley grass extracts treated at different day and night temperatures.

Grass Type	Day and Night Temperature	Phenolic (µg/mL)	Flavonoids (µg/mL)	Vitamin C (µg/mL)
Wheat	10/5 °C	144.15 ± 2.36 a ^z^	47.45 ± 4.29 a	2.45 ± 0.29 a
	20/15 °C	81.76 ± 1.46 b	28.00 ± 3.67 b	1.63 ± 0.15 b
	30/25 °C	97.22 ± 0.52 c	36.18 ± 3.89 ab	1.70 ± 0.03 b
	*p* value	***	*	*
Barley	10/5 °C	92.23 ± 1.73 a ^z^	36.86 ± 4.15 a	1.95 ± 0.04 a
	20/15 °C	75.67 ± 1.03 b	23.55 ± 3.71 ab	0.87 ± 0.02 c
	30/25 °C	80.42 ± 0.86 b	31.73 ± 1.41 b	1.36 ± 0.05 b
	*p* value	***	*	***

^z^ Mean separation within columns according to the Tukey test (*n* = 5 ± standard error). *, *** indicates significance at *p* ≤ 0.05 and 0.001, respectively. Values with the same letters in a column are not significantly different by Tukey test.

**Table 3 foods-10-02742-t003:** Antioxidant enzyme activity of barley and wheat grass extracts treated at different day and night temperatures.

Grass Type	Day and Night Temperature	Guaiacol Peroxidase Activity (POD) **(unit/min/mL)**	Catalase Activity (CAT) **(unit/min/mL)**	Glutathione Reductase (GR) **(unit/min/mL)**	Superoxide Dismutase (SOD)-like Activity (%)
Wheat	10/5 °C	0.10 ± 0.01 a ^z^	14.33 ± 0.59 a	0.18 ± 0.01 a	34.55 ± 0.20 a
	20/15 °C	0.07 ± 0.01 ab	13.21 ± 0.35 ab	0.14 ± 0.01 ab	22.38 ± 1.47 ab
	30/25 °C	0.05 ± 0.01 b	11.28 ± 0.59 b	0.12 ± 0.01 b	15.98 ± 2.91 b
	*p* value	*	*	*	***
Barley	10/5 °C	0.09 ± 0.02 a ^z^	16.07 ± 0.58 a	0.06 ± 0.01 a	12.43 ± 0.92 a
	20/15 °C	0.06 ± 0.01 ab	14.59 ± 0.52 ab	0.04 ± 0.01 ab	9.89 ± 1.03 ab
	30/25 °C	0.04 ± 0.01 b	13.24 ± 0.59 b	0.03 ± 0.01 b	8.20 ± 0.36 b
	*p* value	*	*	*	**

^z^ Mean separation within columns according to the Tukey test (*n* = 5 ± standard error). *, **, *** correspond to significance at *p* ≤ 0.05, 0.01, and 0.001, respectively. Values with the same letters in a column are not significantly different by Tukey test.

**Table 4 foods-10-02742-t004:** Antioxidant activities of young wheat and barley grass extracts treated at different growing conditions.

Grass Type	Day and Night Temperature	2,2′-Azino-bis (3-ethylbenzothiazoline-6-sulphonic Acid (ABTS) (IC_50_, µg/mL)	2,2-Diphenyl-1-picrylhydrazyl (DPPH) (IC_50_, µg/mL)
Wheat	10/5 °C	47.94 ± 0.1 a ^z^	39.26 ± 0.1 a
	20/15 °C	43.64 ± 0.5 ab	33.02 ± 0.0 c
	30/25 °C	45.80 ± 0.1 b	36.53 ± 0.1 b
	*p* value	***	***
Barley	10/5 °C	46.39 ± 0.2 a ^z^	11.70 ± 0.0 a
	20/15 °C	42.46 ± 0.4 ab	8.46 ± 0.0 c
	30/25 °C	43.79 ± 0.1 b	10.68 ± 0.0 b
	*p* value	***	***

^z^ Mean separation within columns according to the Tukey test (*n* = 5 ± standard error). *** corresponds to a significance of *p* ≤ 0.001. The ABTS and DPPH radical scavenging capacities are expressed as the half-maximal inhibitory concentration (IC_50_) equivalent antioxidant capacity (µg/mL). Values with the same letters in a column are not significantly different by Tukey test.

## Data Availability

Not applicable.

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
