# Peer review of "Influence of Temperature Conditions during Growth on Bioactive Compounds and Antioxidant Potential of Wheat and Barley Grasses"

_foods, 2021, doi:10.3390/foods10112742_

Round 1
Reviewer 1 Report
What is new from the above research? Numerous works discuss the effect of temperature on germination and crop growth. The results of this work are predictable from start to finish on the basis of many previous studies on this topic also for wheat and barley. I do not understand what this research brings new to the previous ones. The lack of any significant new information in the work. Moreover, the analysis of compounds by spectrophotometric methods is only an orientation method and cannot prove the value of the raw material. Remaining ambiguities and errors could be corrected.
Line 245-250:” Further, the low and high growth temperatures reduced the chlorophyll contents of the wheat and barley grass extracts. The fixation of carbon dioxide in the Calvin cycle is sensitive to atmospheric stresses, including low and high temperatures, which can decrease photosynthesis in the plant, resulting in a decreased chlorophyll content…”- The effect of temperature on photosynthesis is discussed in biology textbooks for secondary school. This is not revealing.
Line 128: Why do the authors analyzed the anthocyanin content of grasses of wheat and barley? Anthocyanins are plant pigments with a red, purple color and the like. In my opinion, the study on the content of these compounds in the green parts of grasses is unjustified, because their content in these species is very low. Which compounds of this group have been detected in significant amounts in grasses of wheat or barley?
Line 202-203: “whereas the 10/5°C and 30/25 °C (day/night) growth temperature conditions hampered the germination rate” - How? These data are not included in the table and are not discussed in the text.
Line 222: what is: Color value analysis? line 228-229: “The highest greenness (-a*) was observed in the 20/15 °C of the wheat and barley grass extracts, which is probably because of an increased chlorophyll content. or line 232: “Low greenness values indicate carotenoids in plant cells” – After all, the content of carotenoids and chlorophyll was determined in the work, we do not have to conclude by the colors which dyes are more and less. I do not understand the sense of analyzing such parameters in the above work. What measurable value and quality does this analyses testify to, and what would their tangible contribution to the research?
Both analyses of antioxidant activity are based on the same principle - binding free radicals, so it is not surprising that their results are correlated, if polyphenolic compounds are to be responsible for this activity, because they react similarly in the DPPH and ABTS test. Why were these two tests used? Why were the results of these tests not reported as IC50, which is a universal value and makes it possible to relate to the potency of other plant extracts? There is no control in these analyses.
Other problems are less important like for example
- Line 40: why conversely?
- Line 93: what is 5 mL of N, N-dimethylformamide treatment?
- There is a problem with the font format throughout the text.
Author Response
Thanks to reviewer, editor and chief editor for cordial help to improve our manuscript. We have added the reviewer’s comments in the main manuscript and in the author’s response document. The orange color indicated the Reviewer 1, responses. In addition, pink color in the manuscript indicated the author addition.
Reviewer #1
|
Reviewer #1 comments |
Authors’ response |
|
What is new from the above research? Numerous works discuss the effect of temperature on germination and crop growth. The results of this work are predictable from start to finish on the basis of many previous studies on this topic also for wheat and barley. I do not understand what this research brings new to the previous ones. The lack of any significant new information in the work. Moreover, the analysis of compounds by spectrophotometric methods is only an orientation method and cannot prove the value of the raw material. Remaining ambiguities and errors could be corrected. |
Thank you for your valuable comments to improve this manuscript. Despite numerous works on germination and crop growth, specific studies regarding young barley and wheat grass production for food uses are very limited.
As commented, further additional study will be conducted to identify individual compounds using chromatographic methods in the near future.
|
|
Line 245-250:” Further, the low and high growth temperatures reduced the chlorophyll contents of the wheat and barley grass extracts. The fixation of carbon dioxide in the Calvin cycle is sensitive to atmospheric stresses, including low and high temperatures, which can decrease photosynthesis in the plant, resulting in a decreased chlorophyll content…”- The effect of temperature on photosynthesis is discussed in biology textbooks for secondary school. This is not revealing. |
Thank you for your valuable comments to improve this manuscript. |
|
Line 128: Why do the authors analyzed the anthocyanin content of grasses of wheat and barley? Anthocyanins are plant pigments with a red, purple color and the like. In my opinion, the study on the content of these compounds in the green parts of grasses is unjustified, because their content in these species is very low. Which compounds of this group have been detected in significant amounts in grasses of wheat or barley? |
Despite probably low amount, presence of anthocycanins in wheat sprouts or wheat-grass juice was observed in some literature. For instance: Sytar et al., 2018 (doi:10.3390/molecules23092282); Sharma et al., 2020 (doi:10.3390/molecules25245785). Also we have applied 150 µmol photons m-2 s-1 with quantum dot light-emitting diodes in 3 growing conditions to test any changes in grass pigments including anthocyanins. |
|
Line 202-203: “whereas the 10/5°C and 30/25 °C (day/night) growth temperature conditions hampered the germination rate” - How? These data are not included in the table and are not discussed in the text. |
The germination rates of the wheat seeds were 83.75% at 10/5 °C and 75% at 30/25 °C. In barley seeds, the germination rates were 80% at 10/5 °C and 73.75% at 30/25 °C.
|
|
Line 222: what is: Color value analysis? line 228-229: “The highest greenness (-a*) was observed in the 20/15 °C of the wheat and barley grass extracts, which is probably because of an increased chlorophyll content. or line 232: “Low greenness values indicate carotenoids in plant cells” – After all, the content of carotenoids and chlorophyll was determined in the work, we do not have to conclude by the colors which dyes are more and less. I do not understand the sense of analyzing such parameters in the above work. What measurable value and quality does this analyses testify to, and what would their tangible contribution to the research? |
Color value was analyzed to confirm the barley and wheatgrass extract color because the final product was grass juice.
Determining color values can be an indirect indication for pigments present in barley and wheatgrass extract. In this experiment, it was observed that the greenness of barley and wheat grass extract was correlated to chlorophyll content.
|
|
Both analyses of antioxidant activity are based on the same principle - binding free radicals, so it is not surprising that their results are correlated, if polyphenolic compounds are to be responsible for this activity, because they react similarly in the DPPH and ABTS test. Why were these two tests used? Why were the results of these tests not reported as IC50, which is a universal value and makes it possible to relate to the potency of other plant extracts? There is no control in these analyses. |
There are many types of antioxidant assay that are based on different mechanisms, and researchers in many references usually use at least two different assays to measure the antioxant activity of samples. We thought that the ABTS and DPPH assays are commonly applied in many crops to confirm the antioxidant activities.
We have changed to IC50.
|
|
· Line 40: why conversely? |
Thank you. We have discarded conversely. |
|
· Line 93: what is 5 mL of N, N-dimethylformamide treatment? |
The chlorophyll contents of the fresh young wheat and barley grasses (0.10 g) were examined using 5 mL of N, N-dimethylformamide (99.8%) for 24 h. |
|
· There is a problem with the font format throughout the text. |
Thank you. We have formatted the manuscript. |
Reviewer 2 Report
The article is of interest, but it is a bit poor and it would have been desirable to work with different intensities and power of the LED lamps and / or with different extraction methods, etc ...
The article need to make the suggested changes or specifications:
1.- light intensity 150 µmol m-2 s-1 8-days, should say 150 µmol photons m-2 s-1
2.- specify the state of humidity and size of the pieces of the plant before grinding the 10 g of grass and the standardized method followed and compare in results and discussion how the extraction method affects (H2O, hydroalcoholic, milling, microwaves, etc) to the performance of active products.
3.- specify in material and methods where the enzymes and analysis reagents have been acquired. Superoxide dismutase does not show EC number and where it was purchased.
5.- Some photograph of the growth / color / chemical analysis tests carried out and showing the differences observed for the working conditions tested.
Author Response
Thanks to reviewer, editor and chief editor for cordial help to improve our manuscript. We have added the reviewer’s comments in the main manuscript and in the author’s response document. The green color indicated the Reviewer 2, responses. In addition, pink color in the manuscript indicated the author addition.
Reviewer #2
|
Reviewer #2 comments |
Authors’ response |
|
Comments and Suggestions for Authors The article is of interest, but it is a bit poor and it would have been desirable to work with different intensities and power of the LED lamps and / or with different extraction methods, etc ... The article need to make the suggested changes or specifications: 1.- light intensity 150 µmol m-2 s-1 8-days, should say 150 µmol photons m-2 s-1 |
Thank you for your valuable comments.
As commented, further studies can be considered conducting experiments using different LED conditions or different extraction methods. In this study, we used distilled water as a solvent, since we are trying to develop grass juice using barley and wheatgrass
We have changed 150 µmol photons m-2 s-1 |
|
2.- specify the state of humidity and size of the pieces of the plant before grinding the 10 g of grass and the standardized method followed and compare in results and discussion how the extraction method affects (H2O, hydroalcoholic, milling, microwaves, etc) to the performance of active products. |
Approximately 10 g (1 cm length) of each raw grass (separately) was ground with a pestle in 40 mL of dH2O.
In this experiment, we used deionized water extraction as our expected end product would be grass juice. |
|
3.- specify in material and methods where the enzymes and analysis reagents have been acquired. Superoxide dismutase does not show EC number and where it was purchased. |
The enzymes and analysis reagents have been acquired from Sigma-Aldrich, Korea. Superoxide dismutase -like activity (EC 1.15.1.1) |
|
4.- Some photograph of the growth / color / chemical analysis tests carried out and showing the differences observed for the working conditions tested. |
Thank you. We will use later experiment. |
Round 2
Reviewer 1 Report
The authors corrected some minor errors, but did not change the essence of the lack of novelty in the work and the small value of the experiments performed. This paper does not include enough relevant new research. The part of the analysis performed is irrelevant and is only done to increase the volume of the work, but they are not substantiated. This proves that the study was poorly planned.
The answer to some of the comments is unsatisfactory:
- "further additional study will be conducted to identify individual compounds using chromatographic methods in the near future" - it must first be done before it is be possible to publish the research.
At the very least, a quantitative analysis of the compounds present in the extracts should be added and parts, the performance of which is unjustified for the topic of work should be removed.
- "Despite probably low amount, presence of anthocycanins in wheat sprouts or wheat-grass juice was observed in some literature."- even if they have been found, but in small numbers, what is the point of marking them? Even more so with the spectrophotometric method, which is very preliminary and non-specific. They will not matter for the value of the raw material. On this principle, it could also mean, for example, sterols, essential oil or others.
-“ Determining color values can be an indirect indication for pigments present in barley and wheatgrass extract.”- But why do we need an indirect indicator, if we have in the experiment an direct one in the form of an analysis of the content of pigments: chlorophylls or carotenoids, which was determined by the authors. In my opinion, these analyzes made no sense and only serve to increase the volume of manuscript.
-“ We thought that the ABTS and DPPH assays are commonly applied in many crops to confirm the antioxidant activities.”- yes, but these tests have a similar mechanism. While keeping only one of the above free radical tests, it would be worth adding assay based on a different mechanism activity to be able to say something more about the antioxidant potential of the extracts.
There is no control in antioxidant analyses.
Author Response
Thanks to reviewer, editor and chief editor for cordial help to improve our manuscript. We have added the reviewer’s comments in the main manuscript and in the author’s response document. The orange color indicated the Reviewer 1, responses. In addition, pink color in the manuscript indicated the author addition.
Reviewer #1
|
Reviewer #1 comments |
Authors’ response |
|
The answer to some of the comments is unsatisfactory: - "further additional study will be conducted to identify individual compounds using chromatographic methods in the near future" - it must first be done before it is be possible to publish the research. |
Thank you. We have updated this manuscript.
|
|
At the very least, a quantitative analysis of the compounds present in the extracts should be added and parts, the performance of which is unjustified for the topic of work should be removed. - "Despite probably low amount, presence of anthocycanins in wheat sprouts or wheat-grass juice was observed in some literature."- even if they have been found, but in small numbers, what is the point of marking them? Even more so with the spectrophotometric method, which is very preliminary and non-specific. They will not matter for the value of the raw material. On this principle, it could also mean, for example, sterols, essential oil or others. |
Anthocycanins discarded. |
|
-“Determining color values can be an indirect indication for pigments present in barley and wheatgrass extract.”- But why do we need an indirect indicator, if we have in the experiment an direct one in the form of an analysis of the content of pigments: chlorophylls or carotenoids, which was determined by the authors. In my opinion, these analyzes made no sense and only serve to increase the volume of manuscript. |
Color values discarded.
|
|
- “We thought that the ABTS and DPPH assays are commonly applied in many crops to confirm the antioxidant activities.”- yes, but these tests have a similar mechanism. While keeping only one of the above free radical tests, it would be worth adding assay based on a different mechanism activity to be able to say something more about the antioxidant potential of the extracts. There is no control in antioxidant analyses. |
We understand it would be worth adding a different assay. It is not possible to do more study right now. Unfortunately, we used up experimental samples. Also we thought that the ABTS and DPPH assays were enough to test, because lots of papers reported only these two assays in their experiments (doi:10.3390/molecules23092282; doi:10.3390/molecules25245785; doi:10.1007/s11696-017-0288-3; doi: 10.1007/s12161-014-0005-6). α-Tocopherol was used as a reference in this experiment (doi: 10.3746/pnf.2013.18.3.188). |
Reviewer 2 Report
Accept in present form
Author Response
Thanks to reviewer, editor and chief editor for cordial help to improve our manuscript. We have added the reviewer’s comments in the main manuscript and in the author’s response document. The pink color in the manuscript indicated the author addition.
Reviewer #2
|
Reviewer #2 comments |
Authors’ response |
|
Accept in present form |
Thank you to accept in present form. |